# *arfA* antisense RNA regulates MscL excretory activity

Rosa Morra[1],*, Fenryco Pratama[1,2],*, Thomas Butterfield[1], Geizecler Tomazetto[1], Kate Young[1], Ruth Lopez[1], Neil Dixon[1]

Excretion of cytoplasmic protein (ECP) is a commonly observed phenomenon in bacteria, and this partial extracellular localisation of the intracellular proteome has been implicated in a variety of stress response mechanisms. In response to hypoosmotic shock and ribosome stalling in *Escherichia coli*, ECP is dependent upon the presence of the large-conductance mechanosensitive channel and the alternative ribosome–rescue factor A gene products. However, it is not known if a mechanistic link exists between the corresponding genes and the respective stress response pathways. Here, we report that the corresponding *mscL* and *arfA* genes are commonly co-located on the genomes of Gammaproteobacteria and display overlap in their respective 3′ UTR and 3′ CDS. We show this unusual genomic arrangement permits an antisense RNA–mediated regulatory control between *mscL* and *arfA*, and this modulates MscL excretory activity in *E. coli*. These findings highlight a mechanistic link between osmotic, translational stress responses and ECP in *E. coli*, further elucidating the previously unknown regulatory function of *arfA* sRNA.

## Introduction

Bacteria are subject to numerous and occasionally drastic intracellular challenges and extracellular perturbations to which they respond by triggering their defence mechanisms through the expression of stress-responsive genes (1, 2). These mechanisms are necessary for repairing damage, returning cellular homeostasis, and increasing cell survival. One of the common stress conditions that microorganisms encounter is translational stress which can be caused by antibiotics targeting the ribosome and chemical/physical damage of mRNA, resulting in ribosome stalling on nonstop mRNA (3, 4). Furthermore, because of transcription–translation coupling and rapid transcript turnover in bacteria, incomplete transcripts (both nascent and decay intermediates) are known to be ubiquitous, requiring efficient mechanisms to resolve stalled ribosomes (5). Recently, it has also been shown that ribosome collisions lead to recruitment of the endonuclease SmrB that cleaves mRNA and triggers ribosome rescue (6). Bacteria have developed a number of response mechanisms to deal with this translational stress, the primary ribosome rescue system (trans-translation) consisting of the *ssrA* (tmRNA) and SmpB complex mediates the release of the stalled ribosome, targets the truncated peptide for degradation, and promotes mRNA turnover (3, 7, 8). Previous studies have shown that when trans-translation is impaired or overloaded, alternative tmRNA–independent ribosome rescue system(s) prevent translational collapse (9). One such system, the alternative ribosome–rescue factor A (ArfA), is upregulated when the trans-translation system is overloaded. Indeed, *arfA* mRNA undergoes RNaseIII processing to generate a mature transcript lacking a stop codon which in turn is recognised and degraded by the tmRNA and SmpB complex (10, 11). When the trans-translation system is impaired or overloaded, *arfA* transcript escapes tmRNA-mediated degradation allowing active ArfA protein to be produced to provide a backup system for ribosome rescue (7, 11). How the ArfA protein is released from the stalled ribosome, when tmRNA system is compromised, is currently unknown.

Another stress condition that bacteria commonly face is osmotic stress in response to sudden changes in external osmolarity. To counteract the impact of rapid cell volume expansion upon hypoosmotic shock, bacteria use a group of inner membrane proteins known as mechanosensitive (MS) channels (12, 13). In *Escherichia coli*, the two major MS channels are the small-conductance (MscS) and large-conductance (MscL) mechanosensitive channels (14, 15). MscL has a larger channel pore (~30 Å) than MscS (~14 Å) in the open state and requires greater pressure for gating (MscL 10 mN/m, MscS 5 mN/m) (14, 16). Because of the relatively large channel pore size, MscL jettisons not only water and solutes but also cytoplasmic proteins during its gating (17, 18, 19). Interestingly, in *E. coli*, *mscL* and *arfA* are genomically co-located and arranged in a convergent tail-to-tail orientation and share an extended complementary region between the 3′ UTR of *mscL* (20) and 3′ CDS of *arfA*. Furthermore, MscL has been identified as a potential antibacterial drug target and has also been implicated in the cellular uptake of the ribosome-targeting antibiotic streptomycin (21, 22). Gentamycin, another aminoglycoside from this class, and also chloramphenicol have been shown to up-regulate *arfA* transcript levels (23).

We previously discovered that *E. coli* cells encountering both hypoosmotic stress and *arfA*-mediated response to translational

[1]Department of Chemistry, Manchester Institute of Biotechnology, The University of Manchester, Manchester, UK    [2]Institut Teknologi Bandung, Bandung, Indonesia

Correspondence: neil.dixon@manchester.ac.uk
*Rosa Morra and Fenryco Pratama contributed equally to this work

stress undergo a proteome mis-localisation phenomenon, via the MscL channel (17), which we termed excretion of cytoplasmic proteins (ECP). ECP, also referred to as nonclassical secretion or protein moonlighting, has important implications for how bacterial cells interact with their external environment. Indeed, some of these excreted proteins have been reported to mediate host–pathogen interactions (24), biofilm formation (23), suppress macrophage activation (25), and are linked to survival and pathogenicity (26). Yet, the mechanisms of ECP are poorly understood (27), and it was previously unknown how *mscL* and *arfA* collectively mediate ECP in response to stress.

Here, using a combination of gene content and intergenic distance analysis, promoter–reporter assays, quantitative real time PCR (qRT-PCR), targeted gene deletion, and phenotypic analysis under different stress conditions, we sought to explore the degree of genomic conservation of *arfA* and *mscL* across bacteria, and their regulation in the presence of various stress-related factors. Furthermore, we investigated the antisense RNA (asRNA)–mediated mechanism by which *arfA* regulates the excretion activity of MscL in *E. coli*.

## Results

### *mscL* and *arfA* co-localisation is enriched in gammaproteobacteria

Because of the important roles that *mscL* and *arfA* display in responding to osmotic and translational cell stress both individually (3, 28) and collectively (17) and their genomic co-localisation in *E. coli*, we sought to explore the degree to which their presence is conserved across bacterial and archaeal kingdoms. Hidden Markov models (HMMs) of the MscL and ArfA proteins were obtained from the Pfam database (29), and these were used to query NCBI prokaryote representative and reference complete proteomes. A cladogram was constructed using NCBI taxonomic information (Fig 1A). The presence of MscL-coding sequence is shown to be predominately conserved across the bacteria kingdom present in 72% (2,575/3,594) of the genomes queried and was identified in a small subset of archaea genomes (10%, 22/228). In contrast, ArfA-coding sequence is shown to be present in a smaller number of bacterial genomes (10%, N = 375), within these 75% also contain *mscL* (N = 282). Taxonomic analysis of the resultant clustered genomes indicates that the cluster containing only *mscL* (*mscL*-only) is composed of species from across a wide range of phyla present in the dataset (Fig S1). Notably, actinobacteria, bacteroidetes, and firmicutes are highly enriched with *mscL*-only species comprising 93%, 92%, and 77% of the entire phylum, respectively (Table S1). In contrast, the *arfA*-only cluster is almost exclusively composed of Gammaproteobacteria (95%), and the exception to this is a small subset of Betaproteobacteria, including clinically important *N. meningitidis* and *N. gonorrhoeae*, indicating probable horizontal gene transfer (Fig S1). The *mscL–arfA* cluster is almost entirely restricted to Gammaproteobacteria (94%), enriched with species belonging to the Enterobacterales (49%) and Pseudomonadales (28%) orders (Fig 1A). A small number of Betaproteobacteria exceptions were identified within the *mscL–arfA* cluster (N = 14), including a number of *Neisseria*

species. In contrast to the *mscL*-only cluster, which is much more taxonomically diverse, the *mscL–arfA* cluster displays narrower taxa indicative of an earlier ancestry of the *mscL* gene and suggesting that *arfA* was acquired later after the proteobacteria division.

Intriguingly, the intergenic distance of *mscL* and *arfA* is not conserved, and three major sub-clusters can be assigned based on where the two genes are (*i*) co-located and the 3′ CDS of *mscL* is overlapping with the 3′ CDS of *arfA* (overlap), (*ii*) co-located but are not overlapping (proximal), and (*iii*) not co-located (distal) (Fig 1B). The overlap cluster (N = 25) is predominantly composed of bacterial species belonging to the Enterobacteriaceae family (72%), including important clinical and model genera *Salmonella* and *Klebsiella*. The proximal cluster (N = 103) is composed of Enterobacteriaceae (38%), Pectobacteriaceae (19%), and Yersiniaceae (19%) and includes important clinical and model organisms *E. coli* and *Citrobacter koseri*. The distal cluster (N = 154) is predominately composed of Pseudomonadaceae including important clinical and model organisms *Pseudomonas aeruginosa* and *Pseudomonas putida*. As an interesting side observation, we also noticed that in some instances (N = 71) *arfA* is co-located with other genes instead of *mscL*, displaying both CDS overlap (20%) and proximal (80%) arrangements (Table S2). These observations led us to question whether the enriched co-localisation of *mscL* and *arfA* within Enterobacteriaceae genomes has a regulatory function. To probe this, we sought to explore the intergenic regulation between these genes in *E. coli* (Fig 2A).

### *arfA* negatively regulates *mscL* expression

To evaluate whether *arfA* regulates *mscL* expression, we performed transcriptional studies using qRT-PCR to monitor transcript abundance of these genes in *E. coli wt* and gene-specific deleted strains. Carefully designed single-gene deletions were generated to avoid polar effects (30) and direct disruption of the overlapping genes (Fig S2). Analysis of the steady-state transcript level of *mscL* in the *arfA*-deleted *E. coli* strain (Δ*arfA*) indicated an increase of >twofold in the absence of *arfA* (Fig 2B). Upon episomal expression of full-length *arfA* (p*FL*), the *mscL* transcript abundance was restored to *wt* levels (Fig 2B), confirming a role of *arfA* in the negative regulation of *mscL*.

To explore at which level this regulation occurs, promoter–reporter plasmids (P$_{mscL}$-*sfGFP* and P$_{arfA}$-*sfGFP*) were created by fusing the *mscL* or *arfA* promoters (11, 15) with a reporter gene. First, promoter activity of both genes was assessed in a knockout mutant of *rpoS* (Δ*rpoS*) during both exponential and stationary phases (Fig 2C and D). In agreement with previous findings (15, 31), *mscL* expression was associated with the activity of RpoS, as we observed a reduction in *mscL* promoter activity in the Δ*rpoS* strain relative to the *wt* in both exponential (1.5-fold) and stationary phases (4.0-fold) (Fig 2C). Moreover, the enhanced *mscL* promoter activity observed during stationary phase was dependent upon RpoS, with increasing signal only observed in the *wt* and not in the absence of RpoS (*wt* = 2.29-fold increase, Δ*rpoS* = 1.2-fold decrease) (Fig 2C). The results showed only a minor difference in *arfA* promoter activity between *wt* and Δ*rpoS* (Fig 2D). The *arfA* promoter activity was also similar during exponential and stationary phases in both the *wt* and Δ*rpoS* strains. Taken together, these results show that *arfA* promoter activity is

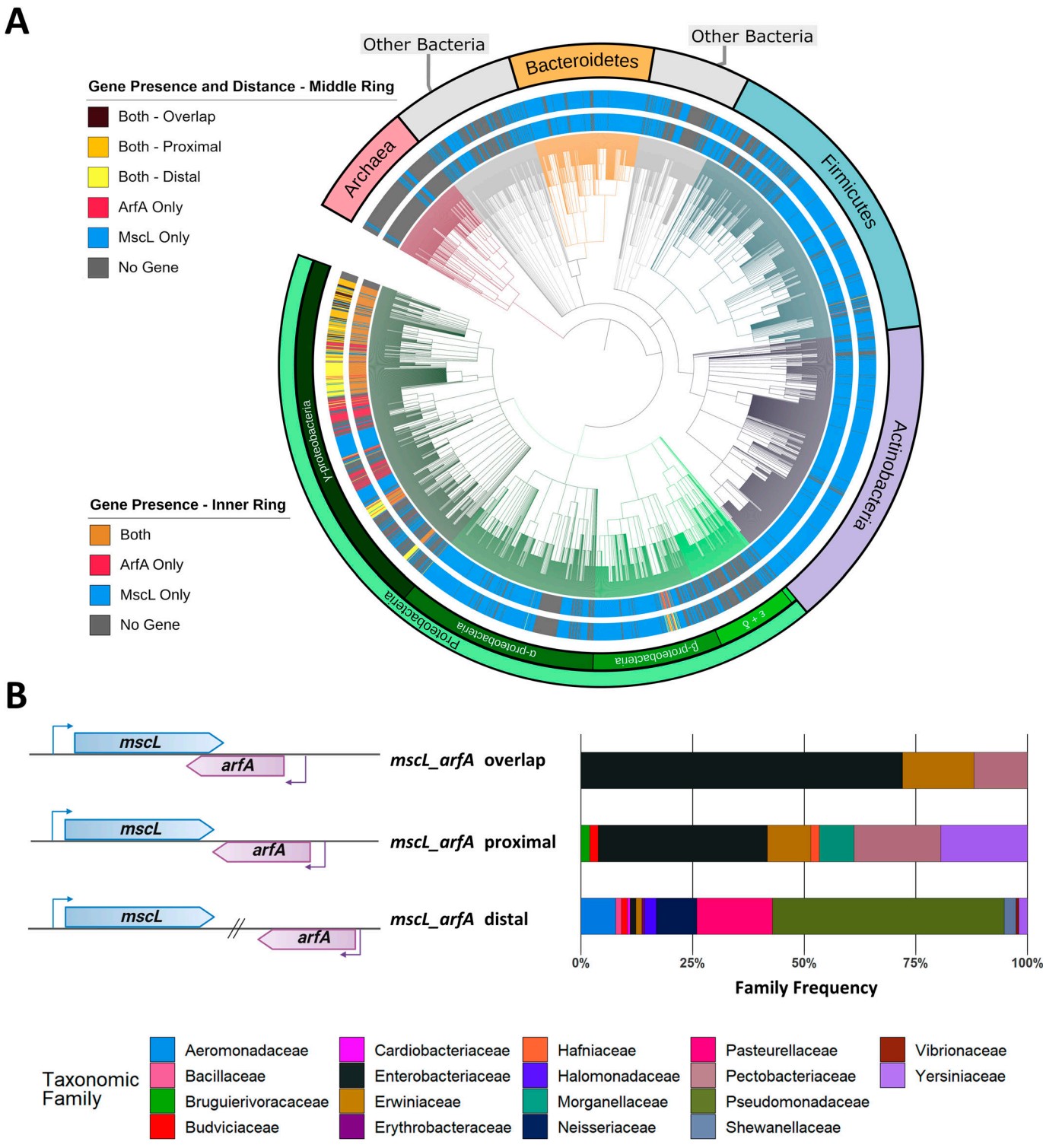

**Figure 1. Taxonomic analysis of *mscL* and *arfA* prokaryotic organisms.**
**(A)** Cladogram of prokaryotic organisms obtained from NCBI representative and reference complete genome/proteome dataset, branches are coloured according to phyla/class, inner-ring indicates presence-absence of *mscL* and *arfA* genes, middle-ring indicates intergenic localisation of *mscL* and *arfA* genes and outer ring indicates taxonomic annotation. **(B)** Genomic co-localization and taxonomic annotation within *mscL–arfA*-containing organisms. The species are clustered according the *mscL* and *arfA* intergenic distance of x nucleotides: overlap (x ≤ 0), proximal (0 < x ≤ 110), and distal (x > 110). See Supplemental Data 1 for genomic and taxonomic data.

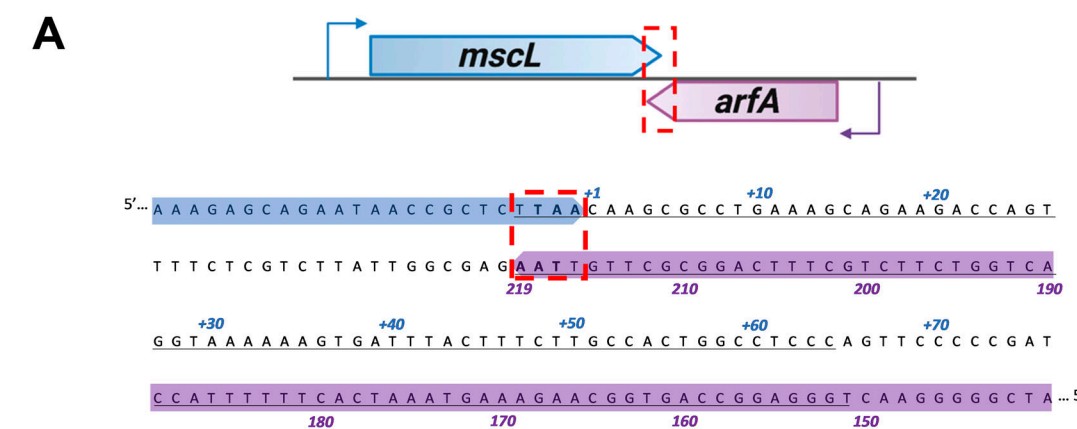

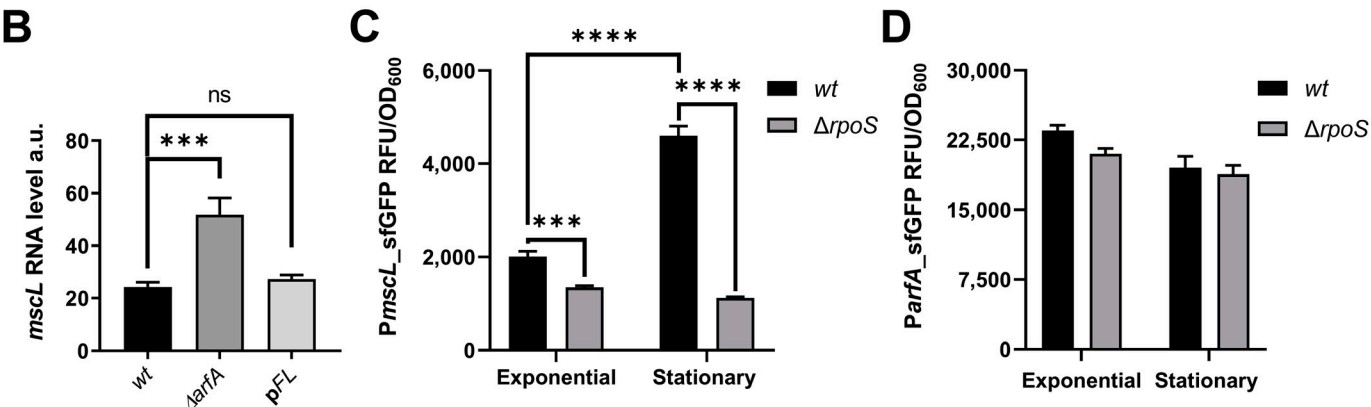

**Figure 2.  *mscL* and *arfA* proximal gene arrangement and expression.**
**(A)** Convergent tail-to-tail gene organisation of *mscL* and *arfA* in the genome of *E. coli*, including the *mscL* 3′ UTR (+1 to +64 nucleotides) and the complementary *arfA* 3′ CDS. **(B)** Steady-state level of *mscL* RNA using qRT-PCR in *wt*, *arfA*-deleted (Δ*arfA*), and full-length *arfA* restored (p*FL*) *E. coli* cells grown in minimal media with an osmolality of 215 mOsm. **(C, D)** *sfGFP* expression driven by P$_{mscL}$ (C) and P$_{arfA}$ (D) in *wt* and Δ*rpoS* strains during cell growth in minimal medium (215 mOsm) at exponential and stationary phases, respectively. The data are shown as relative RNA abundance calculated from Ct values of detected *mscL* normalised to the transcript level of most stable housekeeping genes (see the Materials and Methods section), and as relative fluorescence units (RFU) normalised to OD$_{600}$ (RFU/OD). The error bars represent the SD of at least three biological replicates; multi-comparison ANOVA analysis was performed (P < 0.001***). ns, no significance; a.u., arbitrary units.

between 4 and 10 times greater, and in contrast to the *mscL* promoter is neither significantly affected by growth phase nor by RpoS.

Next, promoter activity was assessed in *arfA* and *mscL* gene–deleted *E. coli* strains, carrying the promoter reporter plasmids, grown in minimal media with low and high osmolality (215 and 764 mOsm) (Fig 3A and B, upper panels, respectively). Consistent with the previous report (15), *mscL* promoter activity in *wt* strain was up-regulated at higher osmolality (~3.5-fold) (Fig 3A upper panel). The pattern was similar in Δ*arfA* and Δ*mscL* strains, indicating that *mscL* promoter activity is independent of the presence of both genes (Fig 3A upper panel). In contrast to *mscL*, the *arfA* promoter activity showed little response to the different osmolalities across all strains (Fig 3B upper panel). These results indicate that *arfA* is not transcriptionally regulated in response to different external osmolalities and that its regulatory function upon *mscL* expression does not occur at the transcriptional level.

Analysis of the *mscL* transcript abundance from the corresponding strains showed an increase in the higher osmolality environment (~1.8-fold, P < 0.0092) (Fig 3A lower panel) consistent to

the increase in promoter activity observed under the same conditions (Fig 3A upper panel). Although both positively regulated in response to osmolality increase, comparison of the *mscL* promoter (3.5-fold) and transcript (1.8-fold) changes suggest that *mscL* is potentially negatively regulated at the post-transcriptional level at high osmolality. In absence of *arfA*, *mscL* transcript abundance also increases independent of the external osmolality (~2.5-fold, P < 0.0004) (Fig 3A lower panel) with no effect observed on the *mscL* promoter activity (Fig 3A upper panel). In contrast, analysis of the *arfA* transcript abundance showed no significant change in *wt* under different osmolalities (Fig 3B lower panel). However, in absence of *mscL*, the *arfA* transcript abundance increased only at high osmolality (2.6-fold, P < 0.0004), whereas no effect upon the *arfA* promoter activity was detected (Fig 3B). Collectively, these results suggest that *arfA* and *mscL* mediate down-regulation of each other at the post-transcriptional level. More specifically, the negative regulation of *arfA* upon *mscL* is observed at both high and low osmolalities, whereas the negative regulation of *mscL* upon *arfA* is only observed at high osmolality.

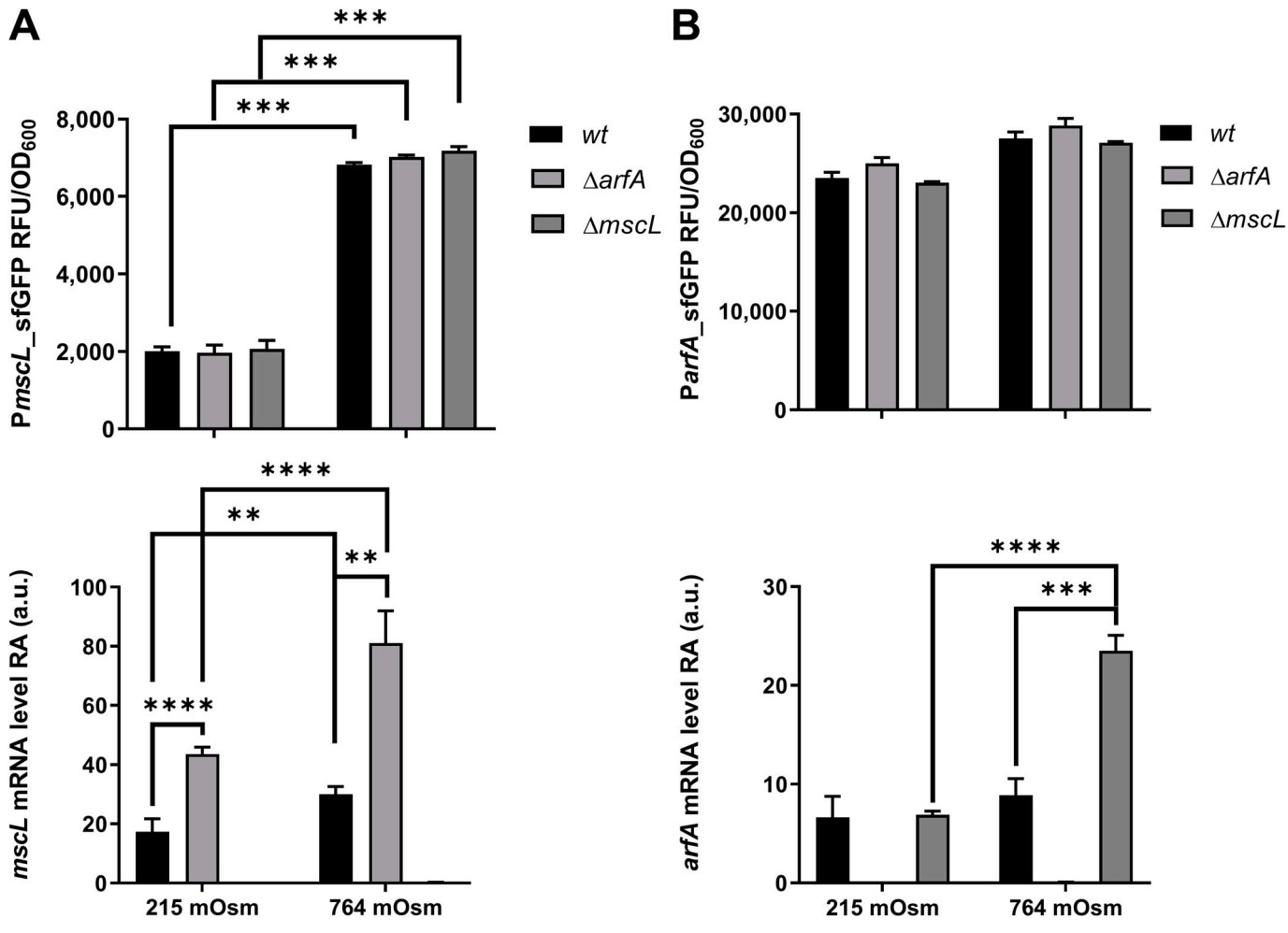

**Figure 3. Transcriptional and post-transcriptional control of *mscL* and *arfA* expression.**
**(A, B)** *sfGFP* expression driven by *mscL* (A upper panel) and *arfA* promoter (B upper panel) and steady-state transcript level of *mscL* (A lower panel) and *arfA* (B lower panel) measured in *wt*, *arfA*, and *mscL*-deleted (Δ*arfA* and Δ*mscL*) *E. coli* strains grown under different osmotic conditions in minimal media (215 and 764 mOsm). Samples were collected during exponential growth. The data are shown as relative fluorescence units (RFU) normalised to OD$_{600}$ (RFU/OD) and as relative RNA abundance calculated from Ct values of detected *mscL* or *arfA* normalised to the transcript level of most stable housekeeping genes. The error bars represent the SD of at least three biological replicates; multi-comparison ANOVA analysis was performed ($P < 0.01$**, $< 0.001$***, $< 0.0001$****). a.u., arbitrary units.

### *arfA* regulates MscL-dependent excretory activity

MscL excretion activity is dependent upon *arfA* in *E. coli* cells that encounter both osmotic and translation stress conditions (17); therefore, we sought to investigate if the MscL excretion activity is regulated by *arfA*. First, cells were grown under high-cell density conditions, where a change in external osmolality is observed between early- and late-stage exponential growth phases, resulting in an osmolality drop of ~70 mOsm (Fig S3A) (17). Consistent with the earlier observation (Fig 3A and B), *mscL* transcript levels were highest in the absence of *arfA* and before the drop in osmolality (Fig S3B and C). This result indicates a synergetic role of *arfA* and osmotic drop in mediating the reduction in *mscL* transcript abundance. Second, to study the impact of *arfA* up-regulation upon *mscL*, we generated a SmpB-deleted strain (Δ*smpB*) in which the primary ribosome rescue system is impaired (11). *arfA* transcript abundance is significantly increased in the Δ*smpB* strain under all

conditions, whereas no change in *mscL* transcript abundance is observed pre-osmolality drop. The greatest reduction in *mscL* transcript abundance was observed in the Δ*smpB* strain post-osmolality drop, providing further evidence of *mscL* down-regulation via a combination of *arfA* and osmolality. Finally, we investigated the phenotypic role (MscL production and excretion) of the *arfA*-mediated down-regulation of *mscL* in cells undergoing both growth-induced osmotic drop and *arfA*-mediated translational stress. To quantify MscL production, we generated *wt* and Δ*arfA* expression strains (BL21(DE3)), encoding a tagged MscL (*mscL::mscLHis*) and bearing an inducible (IPTG) expression plasmid (pET44-sfGFP). The His-tag modification showed no significant impact on both *mscL* expression and MscL excretion activity (Fig S4A and B); and the increase of *arfA* transcript abundance in the presence of IPTG induction confirmed that *gfp* overexpression results in an *arfA*-mediated response to translational stress as previously reported (Fig S4C) (17). Consistent with our previous

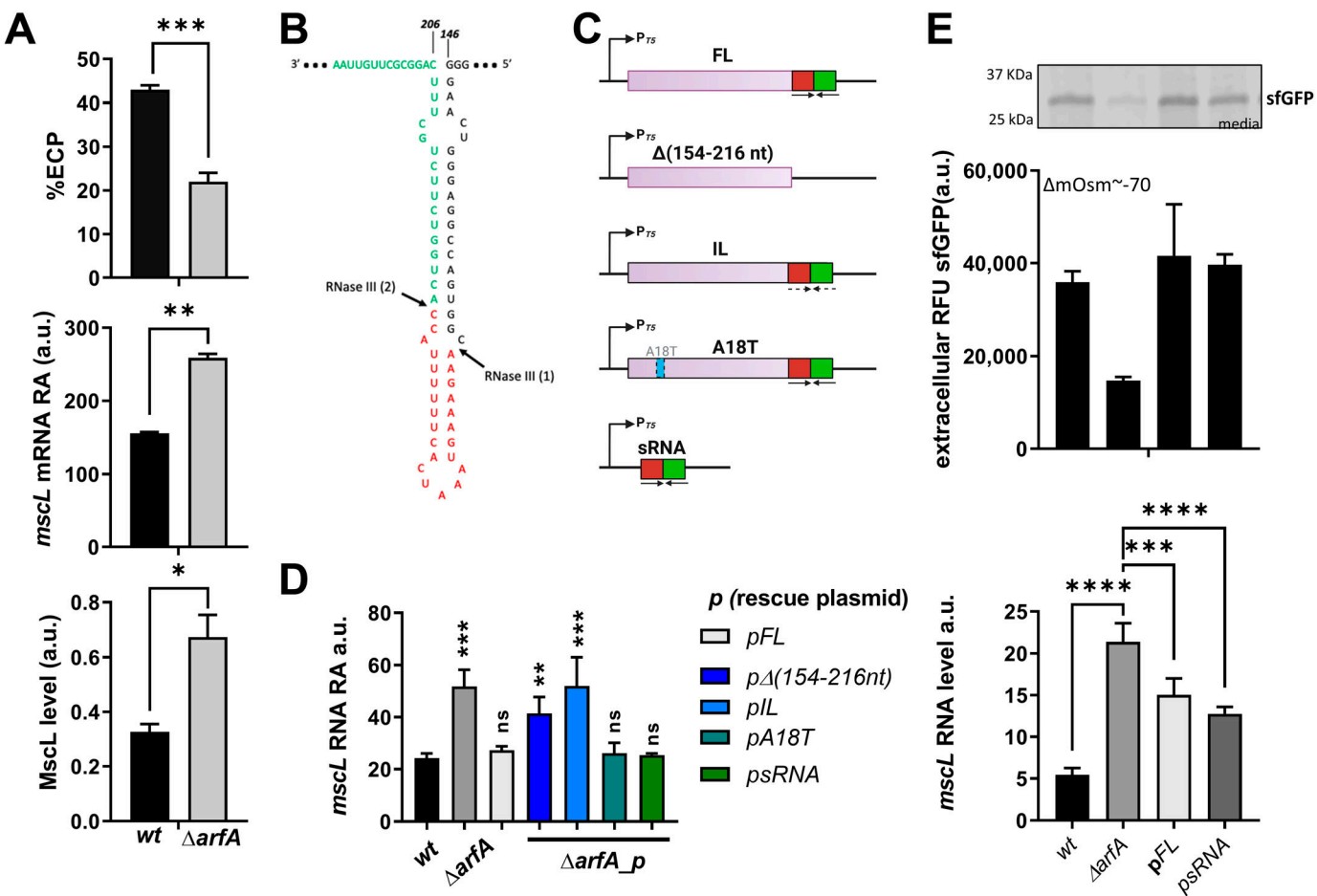

**Figure 4. *mscL* regulation mediated by *arfA* sRNA controls MscL protein level and excretion activity.**
**(A)** Extracellular localisation of sfGFP expressed as %ECP (RFU media/[RFU intracellular+RFU media]) (upper panel), the corresponding *mscL* transcript abundance (middle panel), and the MscL protein levels (lower panel) from *E. coli* BL21(DE3)::*mscLHis wt* and *arfA*-deleted (*ΔarfA*) strains expressing sfGFP, after 15 h of growth in rich media (ΔmOsm ~ −100). MscL Western blot signal was normalised to the signal of RNAP β-polymerase (internal loading control), detected with the anti-β-pol antibody. **(B)** The predicted stem-loop at 3' *arfA* CDS showing the two RNaseIII cut sites at nucleotide position 165 and 189. The *arfA* sRNAs are shown in red and green. **(C, D)** *mscL* transcript abundance in *wt* and *arfA*-deleted (*ΔarfA*) strains bearing an empty pCAN plasmid or one of the constructs in panel (C), grown in minimal media (215 mOsm). **(D)** Analysis of recombinant protein extracellular localisation (upper panel) and *mscL* abundance (lower panel) in *wt* and *arfA*-deleted (*ΔarfA*) *E. coli* BL21(DE3) bearing an empty pCAN plasmid, and *arfA*-deleted *E. coli* BL21(DE3) encoding full-length *arfA* (p*FL*) or *arfA* sRNA (ps*RNA*) during growth in rich media. Samples were collected after osmolality drop (ΔmOsm ~70). A representative SDS–PAGE of the media fractions is shown as inset. **(A, D, E)** The error bars represent the SD of three biological replicates, statistical analysis was performed by unpaired *t* test (A, D), and by one-way ANOVA multi-comparison test (E). ($P < 0.05^*$, $P < 0.01^{**}$, $< 0.001^{***}$, $< 0.0001^{****}$). ns, no significance statistic; a.u., arbitrary units.

finding (17), excretion of cytoplasmic GFP (%ECP) was significantly reduced (≥twofold) in absence of *arfA*, following the drop in media osmolality (Fig 4A upper panel). *mscL* transcript abundance in absence of *arfA* increases (≥twofold) and reflects an increase in MscL protein level (≥twofold) (Fig 4A lower panel). Here, we demonstrate that increased *mscL* transcript level results in increased MscL abundance but counter-intuitively with decreased MscL excretion activity in the absence of *arfA*. Yet, it was previously shown that clustering of MscL, which occurs with increased MscL concentration, promotes channel closure and decreases jettisons activity (32).

### *mscL* transcript is a target of *arfA a*sRNA

The *E. coli* genomic arrangement of *mscL* and *arfA* genes results in an overlapping sequence between the *mscL* 3' UTR and the *arfA* 3'

CDS (Fig 2A). Previous studies (10, 11) identified a regulatory stem-loop structure within the 3' CDS of *arfA* transcript of *E. coli*, from the nucleotide 146 to 206 (Fig 4B). This stem-loop structure contains two sites recognised by RNaseIII that once cleaved produce a truncated transcript without a stop codon and an sRNA 54 nucleotides in length, which can be further processed into sRNAs 24 and 30 nucleotides long (10, 11). To date, the target and function of the *arfA* sRNA was unknown; we therefore investigated if the *arfA* sRNA is involved in the post-transcriptional regulation of *mscL* expression functioning as an antisense RNA. *arfA* gene variants were generated to examine the effect of *arfA* variation upon *mscL* expression. These included: *arfA* full-length (FL); a truncated version missing the regulatory stem-loop (*arfA_Δ*(154-216 nt)) (11); an inverted loop using synonymous codons to maintain correct amino acid coding but disrupt the 3' CDS stem-loop structure (*arfA_IL*) (11);

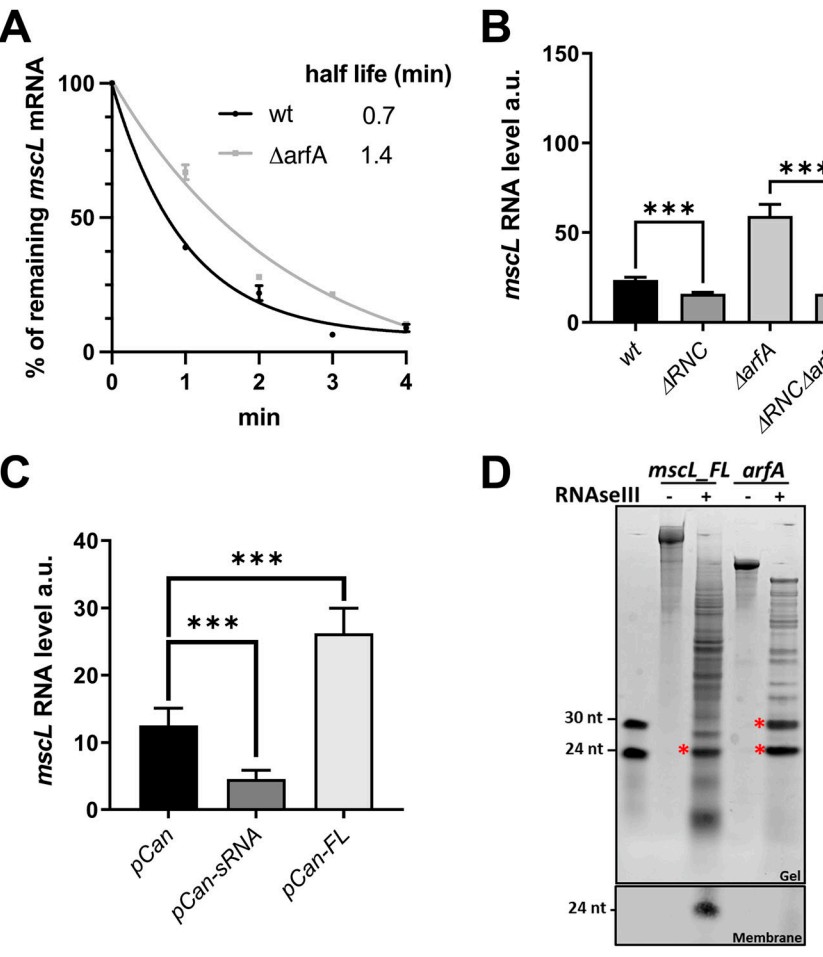

**Figure 5. Effect of *arfA* and RNaseIII upon *mscL* transcript stability and expression.**
**(A)** Effect of *arfA* on the stability of *mscL* RNA determined by plotting the relative abundance of *mscL* versus time post rifampicin treatment. Total RNA was isolated from parental (*wt*) and *arfA*-deleted (Δ*arfA*) *E. coli* K12 cells. The abundance of M1 gene (RNA component of RNaseP) was used as internal standard. Half-life in minutes was determined by one phase decay non-linear fit in GraphPad Prism (version 9). **(B, C)** Steady-state level of *mscL* RNA in a nonfunctional RNaseIII genetic background (Δ*rnc* and Δ*rnc*Δ*arfA*) versus *wt* (B), and in Δ*rnc*Δ*arfA* bearing either empty plasmid (pCan), expressing *arfA* sRNA (pCan-sRNA), or *arfA* FL (pCan-FL) (C). **(D upper panel)** Cleavage assay of in vitro transcribed *mscL_FL* and *arfA* RNAs incubated with *E. coli* RNaseIII. The red asterisks indicate the known sRNAs released by the cleavage of *arfA* and the major cleavage product released from *mscL_FL*. **(D lower panel)** Northern blot analysis of the *mscL_FL* cleavage reaction using 3′ UTR *mscL*-specific biotin labelled probe (spanning from nucleotide at position +13 to +38). IRDye streptavidin antibody was used to detect the signal, and imaging was performing using an Odyssey CLx instrument. Statistical analysis was performed by unpaired *t* test. The error bars represent the SD of minimal three biological replicates. ($P < 0.001$***). a.u., arbitrary units.

a mutant encoding a catalytically inactive version of ArfA (*arfA*_A18T) (33); and the hairpin loop (from nucleotide 160 to 203) (*arfA*_sRNA) (Fig 4C). For context, both *arfA* mutants Δ(154–216 nt) and IL are known to still produce functional ArfA proteins (10, 11). The A18T mutant carries amino acid substitution that has been shown previously to diminish ArfA protein function to rescue stalled ribosomes (33). First, we observed that *arfA* deletion leads to an increase in *mscL* transcript abundance (2.13-fold, $P = 0.0004$) (Fig 4D), consistent with our earlier results (Figs 2B and 3A). Next, we analysed the transcript level of *mscL* in Δ*arfA* strains. In the strains expressing *arfA_FL, arfA_A18T*, and *arfA*_sRNA, *mscL* transcript abundance similar to the *wt* strain was observed. In contrast, in the strains expressing *arfA_Δ(154–216 nt)* and *arfA_IL mscL* transcript abundance was ≅ 2.0-fold higher than *wt*, and similar to Δ*arfA* (Fig 4D). These results show that only those *arfA* gene variants containing the minimal complementary sequence region (nucleotide 160–203) in their 3′ CDS can rescue *mscL* transcript levels, and therefore that this region of *arfA*, and/or the sRNA derived from it, plays a direct role in the negative regulation of *mscL*.

To investigate if the *mscL* regulation mediated by the sRNA of *arfA* has a phenotypic role, we monitored the MscL excretion activity in an *E. coli arfA*–deleted strain expressing *sfGFP* in combination with *arfA*_FL or *arfA*_sRNA undergoing osmotic and translation stress (Fig

4E). Consistent with previous analysis (Fig 4A), *arfA* deletion leads to decreased excretion (twofold, $P = 0.014$) relative to *wt* (Fig 4E upper panel). Intriguingly, *mscL* expression levels observed in the *wt* strain produce the greater ECP activity, whereas the higher *mscL* expression observed in the Δ*arfA* strain leads to reduced ECP (Fig 4E). Upon expression of the *arfA* variants (Δ*arfA*_pFL and Δ*arfA*_psRNA), the excretion phenotype activity was rescued and compared with the *wt* strain. Steady-state analysis of *mscL* transcript levels (Fig 4E lower panel) confirms that the function of *arfA* sRNA is to negatively regulate *mscL* expression at the post-transcriptional level, and in turn to increase MscL excretion activity. Implicitly, this indicates that *arfA* sRNA functions as an antisense RNA (asRNA) and targets the complementary 3′ UTR region of *mscL* mRNA.

### *arfA* asRNA regulates *mscL* transcript stability

Having established that *arfA* asRNA mediates the negative regulation of *mscL* expression, we sought to investigate if it regulates *mscL* transcript stability. We therefore measured the stability of the native *mscL* transcript in the presence (*wt*) and absence of *arfA* (Δ*arfA*). An ~twofold increase in the *mscL* transcript half-life was observed in the absence of *arfA* (Fig 5A). These results support a hypothesis that the observed down-regulation of *mscL* is caused by

formation of a heteroduplex between *arfA* sRNA and *mscL* mRNA which results in a decreased stability of *mscL* transcript. Indeed, antisense mRNA heteroduplexes are known to be cleaved by either endo- (i.e., RNaseIII and RNase E) or exoribonucleases, resulting in the destabilization of the target RNA (34).

### RNaseIII activity plays a role in the post-transcriptional regulation of *mscL*

*arfA* sRNAs are known to be released from the *arfA* transcript via RNaseIII cleavage (10, 11); we therefore next investigated whether RNaseIII affects the *mscL* transcript abundance via *arfA* sRNA. The strain missing a functional RNaseIII protein (RNaseIII(38)) (35) (Δ*rnc*) showed a modest but significant decrease (1.5-fold, *P* = 0.0005) in *mscL* transcript level compared with the *wt* strain (Fig 5B). To further investigate the role of RNaseIII upon *mscL* expression in the absence of *arfA*, we sought to compare *mscL* transcript abundance in absence of both *arfA* and functional RNaseIII (Δ*rnc*Δ*arfA*) and of solely *arfA* (Δ*arfA*). A decrease in *mscL* transcript level (3.7-fold, *P* < 0.0001) was observed in Δ*rnc*Δ*arfA* strain (Fig 5B), suggesting that RNaseIII positively regulates *mscL* expression in an *arfA* independent manner. Comparison of *mscL* transcript abundance between both Δ*rnc* genetic backgrounds (Δ*rnc* and Δ*rnc*Δ*arfA*) show similar levels indicating that unprocessed *arfA*, present in Δ*rnc* strain, is unable to down-regulate *mscL* expression. To validate the latter, we measured the steady-state level of *mscL* transcript in the Δ*rnc*Δ*arfA* genetic background bearing the *arfA*-sRNA or *arfA*-FL plasmid (Fig 5C). We observed a reduction in *mscL* transcript of about threefold (*P* = 0.0007) when *arfA* asRNA was expressed, consistent with the earlier results, confirming the role of *arfA* asRNA in *mscL* regulation (Fig 4E). In contrast, when *arfA*_FL was expressed, no reduction was observed, in fact, *mscL* abundance increased by around twofold (*P* = 0.0003) (Fig 5C), which could be due to the formation of a heteroduplex between *mscL–arfA_FL*. However, *arfA* transcript is believed to predominantly exist in the truncated form in vivo (36), so it is unclear if this heteroduplex is natively found or is an artefact driven by the high levels of *arfA* expression from the rescue plasmid (Fig S5). Overall, these results indicate that RNaseIII has an *arfA*-dependent negative effect and an *arfA*-independent positive effect upon *mscL* expression.

### *mscL* transcript is processed by RNaseIII in vitro

Our results suggest that RNaseIII may also regulate *mscL* expression in an *arfA* independent manner (Fig 5B); intriguingly, Kawano and co-workers identified a 49 nt small asRNA derived from the 3′ UTR of *mscL* (37). We therefore sought to investigate if the stem-loop within the *mscL* 3′ UTR (nucleotide +1 to +63) (Fig S6) is cleaved in vitro by RNaseIII similarly to *arfA* transcript. We synthesised two *mscL* transcript species, one bearing the *mscL* CDS and 3′ UTR stem loop (nucleotide +1 to +70) (*mscL_FL*) and one terminating at the stop codon (*mscL_TAA*). We also synthesised *arfA* transcript as a positive control. As previously reported, we observed that *arfA* transcript is processed by RNaseIII to afford two sRNAs ~24 and ~30 nts, which are derived from the 3′ CDS of *arfA* (nucleotide 166–189 and 190 to stop codon, respectively) (Fig 5D upper panel)

(10, 11). In addition, we observed that *mscL_FL* is also a substrate of RNaseIII in vitro affording ~25 nt sRNA as a major product which is not released from *mscL_TAA* transcript, following the same treatment (Fig 5D upper panel and Fig S7). Furthermore, Northern blot analysis performed with a probe complementary to *mscL* 3′ UTR, spanning between nucleotide +13 and +38, detected a band of ~25 nt further confirming that the RNaseIII cleavage of *mscL* occurs within the 3′ UTR (Fig 5D lower panel). In summary, these results identified *mscL* as an RNaseIII substrate in vitro, which once cleaved releases ~25 nt sRNA from its 3′ UTR, and therefore a truncated *mscL* transcript with a reduced/removed target site for the *arfA* asRNA.

## Discussion

Cellular responses to hyposmotic and ribosome-stalling stress are known to be mediated by the large-conductance mechanosensitive channel (MscL) and alternative ribosome–rescue factor A (ArfA), respectively (3, 13). In addition to osmolytes, MscL has been shown to jettison cytoplasmic proteins (via ECP) (38), which is positively regulated by translational stress response mediated by ArfA (17). Intriguingly, MscL is also implicated in the cellular uptake of streptomycin (21), an aminoglycoside antibiotic that causes miscoding and ribosome stalling (4). In addition, other ribosome-targeting antibiotics have been reported to up-regulate *arfA* (23) and induce ECP (17). More broadly, ECP has been shown to be an important phenomenon associated with a wide variety of biological functions including biofilm formation and host–pathogen interactions (38). Here, in this study, we identified the extent of genomic conservation of *mscL* and *arfA* genes across bacteria and elucidated the mechanism by which these two genes are regulated via intergenic crosstalk and mediate ECP in response to multiple stress stimuli.

It is known that the *mscL* promoter is up-regulated after an increase in external osmolality and upon entry into stationary phase via RpoS (15, 31, 39). In this study, we demonstrate that *mscL* expression has an additional layer of genetic regulation—we discovered an *arfA*-dependent post-transcriptional negative regulation of *mscL*. Furthermore, this newly discovered post-transcriptional down-regulation of *mscL* operates in concert with the known transcriptional regulation of the *mscL* promoter in response to low osmolality. Indeed, when *arfA* is up-regulated, the negative regulation upon *mscL* is only observed post-osmotic drop (Fig S3D). *arfA* transcript is known to be processed by RNaseIII which releases sRNAs with a previously unknown target/regulatory function (10, 36); furthermore, RNaseIII activity has been reported to be positively regulated by hyposmotic shock (40). Here, we identified *mscL* transcript as the target of the *arfA* sRNA, which acts as an antisense RNA to down-regulate *mscL* by reducing the *mscL* transcript half-life (Fig 5A). Importantly, we show the phenotypic function of the *arfA* asRNA is to regulate the MscL excretory activity in response to an osmotic drop (Fig 4E).

Analysis of *mscL* expression under conditions of osmotic and translational stress showed a direct correlation between transcript and protein levels, but surprisingly these were inversely correlated with the excretion activity of MscL (Fig 4A). Indeed, we observed reduced ECP in the Δ*arfA* strain where *mscL* is expressed at a higher

level. MscL channel gating is known to be activated after a decrease in external osmolality, and the resultant lateral force induced turgor pressure within the membrane (15, 41, 42); furthermore, it has been experimentally and computationally shown that MscL protein aggregation effects channel gating (43). Based on the computational modelling of MscL activity, Šarić and co-workers (32) proposed that ungated MscL is present in aggregate cluster-like forms in the membrane during conditions in which MscL is overexpressed, for example, under hyperosmotic condition or in stationary growth phase. This cluster formation promotes channel closure and prevents unwanted gating at low membrane tension. Upon hyposmotic shock, the cell expands and membrane tension increases, leading to separation of the channels from the cluster and individual channel gating. The proposed model shows that MscL clustering is also decreased at lower MscL concentration, and that the pore size of the gated channels depends upon the cluster size and position within it. Previously, it was also shown that MscL channel often passes through sub-states with reduced pore size before opening completely and can also revert to sub-states once opened (44, 45). Considering the results from these previous studies, it is likely that the increased *mscL* expression in the Δ*arfA* strain affects the degree of MscL clustering in the membrane, and that the reduced ECP observed is caused by higher MscL channel numbers in a clustered/ungated and/or partially opened sub-state.

Here, we propose that *arfA* regulates MscL excretion activity during osmotic and translational stress by post-transcriptionally decreasing *mscL* expression via the *arfA* asRNA (Fig 6), in which the *arfA* asRNA attenuates the number of copies of MscL per cell capable of delivering the excretory response upon both translation and osmotic stress stimuli. Analysis of *mscL* transcript abundance in nonfunctional RNaseIII mutant strains (Fig 5B and C) suggests the occurrence of an additional RNaseIII-dependent, but *arfA*-independent, regulation of *mscL*. Furthermore, we demonstrate that in vitro RNaseIII processing of the *mscL* 3′ UTR releases ~25 nt sRNA (Fig 5D). Therefore, we speculate that the RNaseIII processed *mscL* transcript could escape the *arfA* asRNA–mediated down-regulation. Although, it is also possible that the sRNA released from *mscL* 3′ UTR interacts with *arfA* and regulates its transcript processing and/or abundance. We summarise the proposed regulation of *mscL* and *arfA* expression, their crosstalk, and impact on MscL-dependent excretion activity (Fig 6).

To investigate if the presence and genomic co-localization arrangement of *mscL* and *arfA*, as observed in *E. coli*, is evolutionarily conserved, we performed detailed bioinformatic gene content analysis. The data revealed that *mscL* is predominately present across the bacterial kingdom, consistent with previous non-exhaustive analysis (46), but *arfA* appears to be mostly restricted to Gammaproteobacteria. Presumably, the greater occurrence and taxonomic spread of *mscL* is indicative of its physiological importance across a larger array of bacteria and ecological niches relative to *arfA*. It also appears that *arfA* was likely acquired after the Proteobacteria division, as it is not present in Alphaproteobacteria and mostly absent from Betaproteobacteria, apart from a few cases that appear because of horizontal gene transfer. When considering the relevance of the presence/absence of the genes, it is important to note that both MscL and ArfA respond to stresses

that have additional response mechanisms associated with them. In the case of osmotic downshock, a number of other mechano-sensitive channels exist in bacteria that act as a lower gating threshold (14). In turn, ArfA is indeed one of the back-up systems for trans-translation in *E. coli* (33). It is possible therefore that in genomes lacking *mscL* and *arfA* that other genes/mechanisms are present to satisfy the functional requirement of the organism and its ecological niche. Therefore, the question arises as to why certain organisms require additional stress response mechanisms. From the perspective of MscL, it is clear that certain bacterial lifestyles would require greater ability to rapidly response to extreme changes in osmolarity. For ArfA, what drives the requirement for the additional ribosome rescue systems is unclear, some bacteria have additional ribosome rescue mechanisms whereas in others, for example *Mycobacterium tuberculosis*, only trans-translation is present and is therefore essential (47).

In genomes containing both genes, their intergenic distance also varies greatly, and clustering of these indicates enrichment for certain taxa. Genomes in which the genes are co-located (overlap and proximal, N = 128) exclusively belong to species within the order Enterobacterales. In contrast, those genomes in which the genes are distally located (N = 154) are enriched in species belonging to the order Pseudomonadales (52%). Why this variation in intergenic distance occurs is unclear. If *arfA* was indeed acquired subsequently to *mscL*, why has this occurred so differently? Or rather what is the driver for the enriched co-localisation arrangement in these Enterobacterales (observed in ~50% of genomes). In addition, through syntenic analysis of *arfA* genes not co-located with *mscL* (*arfA*-only and distal clusters), we observed alternative genes that similarly displayed an unusual tail-to-tail arrangement with *arfA* overlapping or proximally located CDSs (Table S2). Although further functional analysis is required, based on the *arfA* asRNA regulation upon *mscL* observed here, it appears plausible that additional *arfA*-mediated regulation may be occurring in these alternative genetic contexts. Intriguingly, two of the most abundantly found genes co-located with *arfA* are both associated with iron–sulfur assembly/binding (iron–sulfur cluster assembly protein and 30S ribosomal protein S12 methylthiotransferase), indicating an association with redox sensing and stress. Based on phenotypic and mechanistic analyses demonstrated in *E. coli*, it seems probable that other species with a similar intergenic arrangement (overlap and proximal clusters) may also display an asRNA-mediated cross-regulation between *mscL and arfA*, although this remains to be experimentally validated. However, it has been previously shown that *arfA* transcripts from Gammaproteobacteria are cleaved by RNaseIII (36). In summary, the gene content analysis compliments the biochemical analysis and supports the findings in which this unusual co-localisation of *mscL* and *arfA* is found more broadly than just *E. coli* and most likely has a functional relevance among other Gammaproteobacteria.

This work raises several further important questions related to both mechanistic and physiological relevance. Does the processing of *mscL* 3′ UTR by RNaseIII permit the processed transcript to avoid down-regulation by *arfA* asRNA? Does the rescue of stalled ribosomes and the resultant nucleolytic and proteolytic degradation increase internal cellular osmolality, hyposmotic stress, and therefore RNaseIII activation? There are several reported

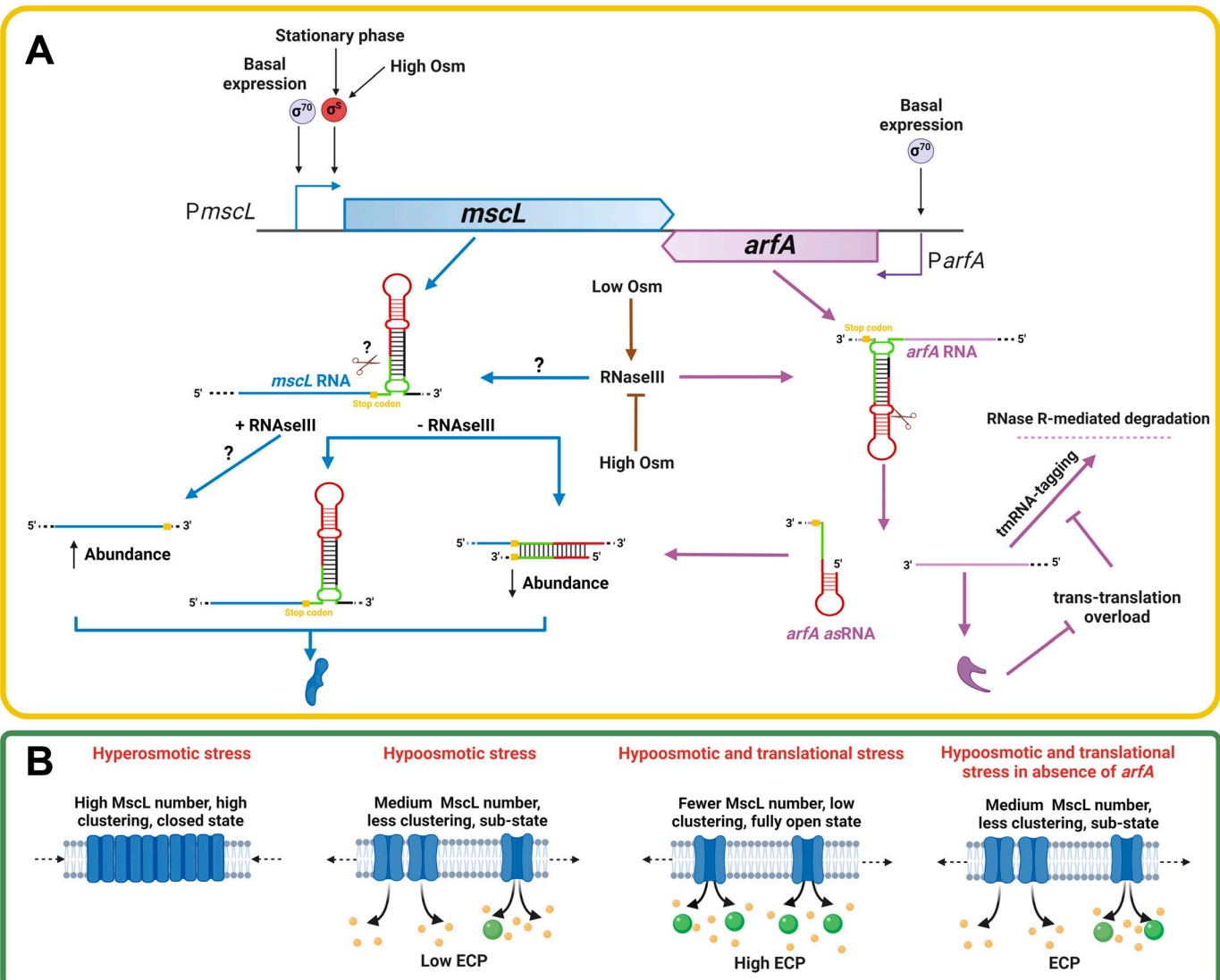

**Figure 6. Proposed regulation of *mscL* and *arfA* expression, antisense *arfA* mechanism, and the impact upon MscL-dependent protein excretion of CPs.**
**(A)** Regulation of *mscL* and *arfA* expression at the transcriptional and post-transcriptional level. *mscL* promoter activity is stimulated under the condition of high external osmolality (Osm), stationary phase growth, and the stress sigma factor ($\sigma^S$/RpoS). *arfA* is constitutively transcribed under the regulation of $\sigma^{70}$/RpoD sigma factor. No evidence of cross-regulation between *mscL* and *arfA* expression at the promoter level was observed in our experiments. *arfA* asRNA is produced upon cleavage of *arfA* transcript by RNaseIII, which is more active under low osmolality (40). *mscL* transcript levels are post-transcriptional down-regulated by the presence of the *arfA* asRNA, resulting in a synergistic effect in combination with the low osmolality. However, RNaseIII processing of *mscL* transcript within the 3′ UTR (demonstrated here in vitro) would avoid this antisense-mediated down-regulation. *arfA* transcript levels are down-regulated by the trans-translation ribosome–rescue system (tmRNA), and up-regulated when tmRNA is overloaded. **(B)** Regulation of MscL gating and excretion activity. MscL channel gating is regulated by lateral force induced by turgor pressure within the membrane (41) and by the quaternary protein aggregation (32). During hyperosmotic stress, MscL protein levels are increased, and MscL channels are present in aggregate/ungated cluster–like forms. Under hyposmotic stress condition, MscL protein levels are decreased, and therefore MscL channels are less clustered/sub-states, and low ECP occurs. During the combination of hyposmotic and translational stress, MscL protein level is further decreased which leads to the increase of MscL channel monomers which are capable of fully gating to enable high levels of ECP. During the combination of hyposmotic and translational stress, but in the absence of ArfA (Δ*arfA*), intermediate numbers of MscL channels are present and moderate ECP occurs.

observations that support our proposed mechanistic link between osmotic and translation stress: (*i*) osmotic challenge leads to cell expansion (hypo) or contraction (hyper), which affects molecular crowding (48), and in turn molecular crowding is known to be important for translation efficiency (49, 50); (*ii*) RNaseIII, which is known to be osmotically regulated, is responsible for processing of ribosomal RNA (51); (*iii*) deletion of ribosome maturation factor, RsgA, has been reported to afford salt tolerance (52); (*iv*) starvation

of *E. coli* affords cross-protection to osmotic challenges (53); and (*v*) pre-treatment with antibiotics targeting translational machinery, impacts cell survival rates after a subsequent osmotic upshift (54) and downshift (55). Therefore, it appears plausible that impairment of the translation apparatus may affect intracellular osmolality.

A secondary set of physiological questions are: (*i*) what is the purpose of ECP when cells experience hyposmotic and translational stress? (*ii*) Does the excretion provide a defence mechanism to

maintain intracellular homeostasis? (*iii*) Does the cell use MscL-mediated excretion following antibiotic-induced ribosome stalling to avoid proteotoxicity? In the primary trans-translation ribosome–rescue pathway, the tmRNA–SmpB complex alleviates ribosome stalling, targets nascent polypeptides for degradation ([3]), and promotes RNA turnover ([8]). In contrast, the alternative ribosome–rescue ArfA pathway displays no degradative activity ([3]). Therefore, it is tempting to speculate that the cell may use MscL excretion activity, regulated by *arfA*, to remove unwanted truncated protein and/or transcripts to prevent proteotoxicity. Finally, why does *E. coli* use antisense post-transcriptional control, in addition to transcriptional control, to down-regulate *mscL* in response to osmotic downshift? If the functional purpose of ECP is indeed to remove truncated protein and/or transcripts, the co-localization and antisense mechanism would permit an energetically efficient means of controlling MscL copy number at the post-transcriptional level, providing a rapid and transient response to osmotic downshift and translation stress. Further investigation will be required to elucidate why the osmotic and translation stress response pathways are coupled, via crosstalk between *mscL* and *arfA*, whether ECP does indeed prevent proteotoxicity following ArfA-mediated ribosome rescue, and whether the *mscL–arfA* regulatory mechanism is evolutionarily conserved in some or all genomic contexts.

# Materials and Methods

### Gene content analysis

3,822 prokaryotic proteomes were recovered from NCBI on 15/3/22. These translated CDS proteomes were recovered based on the criteria that they were first from prokaryota and second were listed as coming from either a reference or representative genome that was fully assembled to the "complete genome" level. These proteomes were filtered to remove any proteins tagged "pseudo" to remove proteins marked as problematic by the NCBI Prokaryotic Genome Annotation Process. HMMs corresponding to the ArfA protein (PF03889.16) and MscL protein (9PF01741.21) were recovered from Pfam ([29]) on 28/3/22 (ArfA) and 31/3/22 (MscL), respectively. HMMER (version 3.1b2) ([http://hmmer.org/](http://hmmer.org/)) was used to search for matches to the two HMMs within the collected proteomes. Hmmsearch was run using the gathering bit scores (25.5 for ArfA and 27.4 for MscL) provided by the HMMs to set model-specific score thresholds and the default values for all other settings (Supplemental Data 1). A Bonferroni corrected *P*-value was calculated by dividing the HMMER calculated *P*-value by two.

### Intergenic distance analysis

Python ([56]) and several packages (NumPy ([57]), pandas ([58]), Biopython ([59]), and SciPy ([60])) were used to calculate the distance between instances of the two genes found in the same genome. Distances were calculated from the last nucleotide of one CDS to the last nucleotide of the other CDS taking the minimum possible intergenic distance assuming genome circularity. Genomes were binned according to which genes they had (*arfA* only, *mscL* only,

both genes, and neither of the genes) and then sub-binned within the "both genes" bin according to distance between the genes. Genomes in which the genes overlapped with one another were labelled "overlap," those which were non-overlapping and closer than or at a distance of 110 nt were labelled "proximal," and those which were also non-overlapping, but further away than 110 nt were labelled "distal." In one case, there was a genome with one gene on a chromosome and the other on a plasmid. This genome was labelled as "distal" (Supplemental Data 1). Metadata and taxonomic information were retrieved for each of the genomes from NCBI on 12/4/22 (metadata) and 23/6/22 (taxonomic information) (Supplemental Data 1). Data were graphed using R, tidyverse ([61]), and Polychrome ([62]) packages. A cladogram of the genomes that we surveyed was produced from the NCBI taxonomy database using phyloT ([https://phylot.biobyte.de](https://phylot.biobyte.de)). This was then visualised using iTOL ([63]).

### Bacterial strains, plasmids, and media

*E. coli* strains and plasmids used in this study are listed in Table S3. To generate target gene deletions in the strain backgrounds MG1655 and BL21(DE3), lambda Red recombineering strategy was performed as described before ([64]), using pRL128 ([65]) and pSIM18 ([66]) plasmids. The primers used to generate FRT-flanked kanamycin selection cassettes and to construct the rescue plasmids are listed in Tables S4 and S3. For cloning purposes and general overnight starter culture, cells were grown in the LB medium (0.5% yeast extract, 0.5% NaCl, 1.0% bactotryptone). For gene expression study in minimal media, cells were grown in citrate–phosphate-defined medium, pH 7, per litre contains 8.58 g Na$_2$HPO$_4$, 0.87 g K$_2$HPO$_4$, 1.34 g citric acid, 1.0 g (NH4)$_2$SO$_4$, 0.001 g thiamine, 0.1 g MgSO$_4$.7H$_2$O, and 0.002 g (NH$_4$)$_2$SO$_4$.FeSO$_4$.6H$_2$O, with 0.04 (starter culture) or 0.2% of glucose (experimental culture), and supplemented with 0 or 0.3 M NaCl (media osmolality of ~215 and 764 mOsm, respectively). For gene expression studies in high cell density culture, cells were grown in TB medium (2.7% yeast extract, 4.5% glycerol, 1.3% bactotryptone, 0.2% glucose). For gene expression study in cells missing a functional RNaseIII protein (Δ*rnc*), cells were grown in LB medium (10 g/liter NaCl, 5 g/l yeast extract, 10 g/liter tryptone).

### Construction of promoter–reporter plasmids

P$_{mscL}$-sfGFP and P$_{arfA}$-sfGFP fragments were synthesised as gBlocks (Integrated DNA Technologies). P$_{mscL}$-sfGFP contained a 321-bp fragment of the *mscL* promoter, including the first 11 codons of *mscL* ([15]). P$_{arfA}$-sfGFP contained *arfA* promoter referred to the sequence from Chadani et al ([11]). The sfGFP CDS is located after the promoter. The DNA fragments were cloned into the low-copy p131B plasmid using NEBBuilder HiFi DNA Assembly following the manufacture's protocol, and the resultant plasmids were used for promoter–reporter activity study.

### Construction of rescue plasmids

All *arfA* gene variant fragments were synthesised as gBlocks (Integrated DNA Technologies) and contained upstream the T5 promoter sequence and downstream the rrnB T1, T7, and lambda T0 terminator sequences. The variant *arfA*_Δ(154-216 nt) contained a

truncated version of full-length gene missing nucleotide 154–216 nt. The variant arfA_IL contained inverted repeats: nucleotide position at 147 (G), 156 (G), and 159 (C) were substituted with A; nucleotide position at 162 (T) was substituted with C; and nucleotide position at 171 (A) was substituted with G. The variant arfA_A18T contained the substitution of the alanine amino acid codon (AGC) in position 18 with threonine (ACC). The variant arfA_sRNA contained only the sequence from nucleotide 160 to 203. The DNA fragments were cloned into the pCA24N plasmid using NEBBuilder HiFi DNA Assembly following the manufacture's protocol, and the resultant plasmids were used for experiments in ΔarfA-rescued strains.

## Growth conditions in minimal media

Cells were first inoculated in 10 ml of citrate–phosphate medium (50 ml falcon tube) supplemented with 0.04% glucose, 50 $\mu$g/ml carbenicillin (plasmid maintenance), and when required 50 $\mu$g/ml kanamycin, then grown overnight 12–16 h at 37°C 200 rpm. The following morning, cultures were supplemented with 0.2% glucose. After one doubling, cultures were diluted 20-fold into fresh citrate–phosphate media supplemented with 0 and 0.3 M NaCl (215 and 764 mOsm, respectively), 0.2% glucose, and 50 $\mu$g/ml carbenicillin, and continued to grow at 37°C at 200 rpm (New Brunswick Scientific I26 shaker) to reach exponential growth phase (4 h) or stationary growth phase (O/N). Samples were collected at indicated time points for further analysis.

## Growth conditions for high-density culture

High-density cultures were obtained by growing the cells in TB medium with an osmolality of ~350 mOsm. After an overnight pre-inoculum in TB, cultures were diluted to $OD_{600}$ 0.02 in 30 ml of fresh TB medium supplemented with 0.2% glucose (UY flask), and then continued to grow at 37°C at 200 rpm (New Brunswick Scientific I26 shaker) until the desired $OD_{600}$ was reached. When required, induction of recombinant protein (sfGFP) was performed by adding 250 $\mu$M IPTG at $OD_{600}$ ~1. Under these conditions, the cultures can grow to an $OD_{600}$ up to ~25 and undergo a growth-induced drop in media osmolality.

## Media osmolality measurement

Cultures were centrifuged (17,000 $g$, 1 min or 5,000 $g$, 10 min) to separate media fraction from the cell pellet. The osmolality of the media fraction was measured with a cryoscopic osmometer (Osmomat 030; Gonotec) following the manufacture's protocol.

## GFP expression and ECP analysis

For promoter–reporter activity assay, GFP expressions from the intact cell were measured. Cultures were harvested (5,000$g$, 10 min) and washed twice in an equal volume of 1 × PBS. Relative fluorescence units (RFUs) and $OD_{600}$ were measured, and GFP expressions were presented as RFU normalised to OD (RFU/OD). For ECP analysis, GFP expressions from the media and cell fractions (periplasm and spheroplast) obtained as previously described (17) were measured directly as RFU. The extracellular localisation of GFP (ECP) was expressed as %ECP (RFU in the medium/RFU intracellular + RFU media). A BMG CLARIOstar microplate reader was used to measure the colorimetric fluorescence and cell density ($OD_{600}$) of intact cells.

## RNA analysis by qRT-PCR

Samples for RNA analysis were collected from ~$10^9$ cells ($OD_{600}$ of 1) grown in minimal or rich media. Cellular RNA was stabilised using RNAlater (Invitrogen) following the manufacture's protocol. RNA extraction was performed following the RNeasy mini protocol (QIAGEN). For quality control of extracted RNA, RNA integrity (RIN) was assessed using the Agilent 2100 Bioanalyser (Agilent Technologies) following the preparation protocol from RNA 6000 Nano kit (Agilent Technologies). Only RNA with a RIN > 7 (67) were used to generate the cDNA by the reverse transcriptase PCR. The RNA samples were snap-frozen in liquid nitrogen and then stored at –80°C. cDNA synthesis was performed following the SuperScript IV VILO protocol (Invitrogen), using ~0.7 $\mu$g total extracted cellular RNA as a template. No RT enzyme control was also included to assess the presence of contaminating genomic DNA. The cDNA was stored for the short term at –20°C and long term at –80°C. The qRT-PCR reaction was performed in a volume of 10 $\mu$l in 96-well clear plates sealed with adhesive (MicroAmp Applied Biosystems) following the SYBR Green Master Mix manufacturer's protocol (Thermo Fisher Scientific). In brief, each 10 $\mu$l reaction, including the no RT control for genomic DNA contamination, comprises 2 $\mu$l of 5× diluted cDNA, a final concentration of 1 × Syber mix (Thermo Fisher Scientific), and 300 nM forward and reverse primers (Table S5). Optimal gene–specific primers (Table S4) were designed using the Roche Online Assay Design Centre and submitted to the Basic Local Alignment Search Tool (BLAST) to check for the nonspecific binding. To compensate for error between samples (variations in cell number, RNA isolation, reverse transcription, etc), two endogenous "house-keeping" transcripts were chosen using the GeNorm algorithm (68). Initially, four reference genes used in previous analysis (recA, idnT, ffh, and cysG) (69, 70) were assayed under the experimental conditions used for this study; cysG and recA, whose expression did not change across the assayed conditions, were selected as the two most stable transcript. In the case of analysis in the nonfunctional RNaseIII strain, cysG was unstable and therefore recA was used for normalisation. Samples were amplified on a QuantStudio3 real-time PCR machine, and the temperature programme was set at 50°C for 2 min and then 95°C for 2 min followed by 40 cycles of 95°C for 15 s and 57°C for 15 s. Two to three technical and at least three biological replicates were performed for each sample. Transcript abundance of target genes expressed in a sample were normalised to the geometric mean of the two endogenous control genes. This is given by $\Delta C_t$, where $\Delta C_t$ is determined by subtracting the geometric mean of endogenous genes $C_t$ value from the average of target gene $C_t$ value ($C_t$ GOI–$C_t$ Ref). Relative abundance (RA) was calculated by the following formula, RA = 100 × ($E$ ^ -$\Delta C_t$), where E stands for amplification efficiency, as determined from the slope of the standard curve of each pair primers ($E = 10^{(-1/slope)}$). Only pair primers with a % E value (= −1 + 10(−1/slope) × 100) ranging between 90% and 110% were used in this study.

## RNA decay analysis

To determine the stability of the *mscL* transcripts, cells were grown in LB medium at 37°C at 200 rpm (New Brunswick Scientific I26 shaker) until $OD_{600}$ ~0.7, at which rifampicin (1 mg/ml) was added to block transcription. Samples were then collected 0,1,2,3, and 4 min after rifampicin addition for analysis of RNA decay by qRT-PCR as described above. The stability of *mscL* RNA was determined by plotting the relative amount of *mscL* versus the time, with the amount of *mscL* RNA at time 0 set to 100%. The relative mRNA abundance was calculated from $C_t$ values of detected *mscL* normalised to the transcript level of the stable M1 gene (RNA component of RNaseP) [71]. Half-life in minutes was determined by one-phase decay nonlinear fit in GraphPad Prism (version 9).

## Western blot

Samples for protein analysis by Western blot were collected from cell cultures normalised to a volumetric $OD_{600}$ of 10. Cultures were centrifuged (5,000 *g*, 10 min), and pellet cells were then resuspended in 400 $\mu$l of lysis buffer (PBS 1× supplemented with Roche protease inhibitor and benzonase). Cells were lysed by sonication (30% Amp, 7 s "ON" 7 s "OFF," cold condition). Lysates were mixed with 2 × SDS–PAGE loading buffer (1:1), and then were boiled for 10 min. Equal amounts of samples were separated by SDS–PAGE and then transferred onto PVDF 0.2 $\mu$M membrane using Trans-Blot Turbo apparatus (Bio-Rad). The membrane was blocked with 5% (wt/vol) skim milk in PBS 1 × (30 min, 50 rpm, room temperature), and then probed with primary antibody (mouse monoclonal anti-His from Pierce, 1:3,000; rabbit monoclonal anti-$\beta$-Pol from Abcam, 1:2,000) overnight, 90 rpm at 4°C. After washes with 1× PBS (three times), the membrane was then incubated with secondary antibody (anti-mouse and anti-rabbit 1:30,000), 30 min, 90 rpm, at room temperature. The protein signals were detected with a LI-COR Odyssey scanner and quantified based on densitometry analysis using LI-COR Image Studio 5.0. All data were measured in biological triplicate.

## In vitro transcription

DNA templates containing *mscL* sequence spanning from the transcription starting site to the 3'UTR (70 nts long) (*mscL_FL*) or to the stop codon (*mscL_TAA*) and *arfA* sequence spanning from the transcription starting size to the stop codon, both bearing the T7 promoter sequence upstream, were synthesized as gBlocks (Integrated DNA Technologies). Transcription of RNAs in vitro by T7 RNA polymerase from the DNA templates was then carried out using the MEGAscript T7 kit (Ambion) according to the manufacturer's instructions. RNAs were then checked for quality by electrophoresis on denaturing urea gel (10% TBU, Novex).

## In vitro RNaseIII cleavage assay

In vitro transcribed RNAs were subjected to RNaseIII cleavage assay as follow: 10 pmol of RNA were used in 20 $\mu$l reaction volume containing 1× enzyme buffer (AM2290; Ambion). Reactions were pre-incubated at 37°C for 5 min, followed by the addition of 0.5 U of *E. coli* RNaseIII (AM2290; Ambion) and further incubation at 37°C for up to

15 min. Reactions were stopped by the addition 5 mM EDTA and purified with ZipTip C18 (10 $\mu$l bed; Millipore). Cleavage products were separated on denaturing urea gel (15% TBU; Novex) which was then stained with 1× SYBRGold and exposed to a PhosphorImager screen.

## Northern blot

The Northern blot was performed according the "Northern blot analysis using bitoin PCR–labelled probes" protocol from LI-COR Odyssey website with few modifications detailed below. The *mscL_UTR* biotin–labelled probe, containing nucleotide +13 to +38 after the *mscL* stop codon, was synthesized as 5'/52-Bio/modified DNA oligo (5'/52-Bio/CACTTTTTTACCACTGGTCTTCTGCT) (IDT). The transfer of RNA from denaturing urea gel onto nylon membrane was performed at 1.0 A constant for 15' using a Trans-Blot Turbo apparatus (Bio-Rad). EDC (E7750; Sigma-Aldrich) was used to cross-link the RNA to the nylon membrane as described previously [72].

# Data Availability

All data generated or analysed in this study are included in this manuscript or in supplemental material.

# Supplementary Information

# Acknowledgements

R Morra, T Butterfield, and R Lopez were supported by BBSRC Responsive mode grant (BB/T005742/1). F Pratama received a PhD scholarship from LPDP Indonesia Endowment Fund for Education (S-903/LPDP.3/2016). G Tomazetto was supported by a UUKI/BEIS Rutherford Fund Strategic Partner (RF-2018-41). N Dixon and K Young were supported by a BBSRC David Phillips Fellowship (BB/K014773/1).

## Author Contributions

R Morra: conceptualization, formal analysis, investigation, methodology, and writing—original draft.
F Pratama: formal analysis, investigation, and writing—original draft.
T Butterfield: resources, data curation, formal analysis, and visualization.
G Tomazetto: resources and data curation.
K Young and R Lopez: investigation.
N Dixon: conceptualization, supervision, funding acquisition, methodology, and writing—review and editing.

## Conflict of Interest Statement

The authors declare that they have no conflict of interest.

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
