## [Reviewer comments · Life Science Alliance]

Life Science Alliance

arfA antisense RNA regulates MscL excretory activity

Rosa Morra, fenryco Pratama, Thomas Butterfield, Geizecler Tomazetto, Kate Young, Ruth Lopez Perez, and Neil Dixon

DOI: <https://doi.org/10.26508/lsa.202301954>

Corresponding author(s): Neil Dixon, University of Manchester

Review Timeline:

Submission Date:	2023-01-27
Editorial Decision:	2023-02-20
Revision Received:	2023-03-02
Editorial Decision:	2023-03-15
Revision Received:	2023-03-20
Accepted:	2023-03-20

Scientific Editor: Novella Guidi

Transaction Report:

February 20, 2023

Re: Life Science Alliance manuscript #LSA-2023-01954-T

Dr. Neil Dixon
University of Manchester
Manchester Institute of Biotechnology
School of Chemistry
131 Princess Street
Manchester M1 7DN
United Kingdom

Dear Dr. Dixon,

Thank you for submitting your manuscript entitled "arfA antisense RNA regulates MscL excretory activity" to Life Science Alliance. The manuscript was assessed by expert reviewers, whose comments are appended to this letter. We invite you to submit a revised manuscript addressing the Reviewer comments.

Thank you for this interesting contribution to Life Science Alliance. We are looking forward to receiving your revised manuscript.

Sincerely,

B. MANUSCRIPT ORGANIZATION AND FORMATTING:

Reviewer #1 (Comments to the Authors (Required)):

The paper studies the interplay between arfA and MscL. The experiments are well controlled, and the conclusions reasonable. I may not agree with all of the speculations in the discussion, but I admit they are not unreasonable if you believe the papers they cite. My only comment is minor. In all of the figures showing pFL data, the bar cannot be seen on any of my computers. The authors need to either change its color or encase it in a line so the boundaries can be seen.

Reviewer #2 (Comments to the Authors (Required)):

Comments on the manuscript "arfA antisense RNA regulates MscL excretory activity"

Based on the observations of Morra et al. 2018 the molecular details of ArfA and MscL regulation were scrutinized. The conspicuous gene arrangement with overlapping 3' UTRs in E. coli was mainly found in Enterobacteria. The asRNA of arfA turned out to be the basis for the complex regulation of the excretion function of MscL under hypoosmotic stress and ribosome stalling conditions. A very careful description of a large amount of work is given. Material and methods are carefully described.

Minor Comments

Are there experiments which demonstrate the fitness increase for E. coli to use this complex interaction between arfA and mscL?

Lines 48 to 51: This part is difficult to understand for an uninitiated reader. It may be mentioned that the release of ArfA from the ribosome in a tmRNA deficient strain is still an unsolved riddle or?

The reviewer is not trained in reading cladograms. It took some time for me to understand that the inner and the outer ring are the two small rings between the circularized tree and the real outer ring describing the different bacterial groups.

Line 226, here it would be helpful to mention that it is known that clustering of MscL reduces its ECP activity.

In Fig. 6 A and B the lettering is rather small - a bit larger would be helpful.

Line 473: Has it been checked if asRNA also occurs in more distantly located mscL and arfA genes?

Reviewer #1 (Comments to the Authors (Required)):

My only comment is minor. In all of the figures showing pFL data, the bar cannot be seen on any of my computers. The authors need to either change its color or encase it in a line so the boundaries can be seen.

Thank you for highlighting this issue; in all the pictures, a black border has been added to the bar graphs

Reviewer #2 (Comments to the Authors (Required)):

A very careful description of a large amount of work is given. Material and methods are carefully described.

Minor Comments

Are there experiments which demonstrate the fitness increase for *E. coli* to use this complex interaction between *arfA* and *mscL*?

Why certain gamma-proteobacteria species have co-located *arfA* and *mscL* genes, whilst others do not, is an interesting evolutionary question i.e. what is the benefit/gain of function of a particular intergenomic arrangement. However, we did not perform any specific experiments to demonstrate this to date. From an experimental design perspective, the assessment of fitness would be difficult to test, as would require identification of two other whilst isogenic species, in which the two genes are either co-located or distally located, to afford a direct fair side-by-side comparison of fitness. However, we do note that previously we demonstrated that an *E. coli* KO of *mscL* and subsequent episomal replacement of *mscL* (i.e. trans) both afforded the same viability (CFU/ml) as the wt strain (Fig3D, Morra et al mBio 2018).

Lines 48 to 51: This part is difficult to understand for an uninitiated reader. It may be mentioned that the release of ArfA from the ribosome in a tmRNA deficient strain is still an unsolved riddle or?

Lines 48 to 51 have been changed to: "...and degraded by the tmRNA and SmpB complex (10,11). When the trans-translation system is impaired or overloaded, *arfA* transcript escapes tmRNA mediated degradation allowing active ArfA protein to be produced to provide a backup system for ribosome rescue (7,11). How the ArfA protein is released from the stalled ribosome, when tmRNA system is compromised, is currently unknown.

The reviewer is not trained in reading cladograms. It took some time for me to understand that the inner and the outer ring are the two small rings between the circularized tree and the real outer ring describing the different bacterial groups.

The two small rings have now been named inner and middle ring, whilst the ring describing the different bacterial groups as outer ring.

Line 226, here it would be helpful to mention that it is known that clustering of MscL reduces its ECP activity.

Text added to line 200-202 in the revised MS “Yet, it was previously shown that clustering of MscL which occurs with increased MscL concentration, promotes channel closure and decreases jettisons activity (32).”

In Fig. 6 A and B the lettering is rather small - a bit larger would be helpful.

The size font of the text has been increased

Line 473: Has it been checked if asRNA also occurs in more distantly located *mscL* and *arfA* genes?

We have not assessed if the *arfA* asRNA is produced from distally located genes. We did perform preliminary bioinformatic analysis to explore the degree of complementary between the genes from proximal and distal group, however the analysis was inconclusive. Firstly, it was difficult to accurately determine the respective transcript stop sites, this was compounded by the fact that *arfA* contains a terminator-like within its CDS. In addition, whether the corresponding transcripts are processed by RNaseIII are difficult to predict computationally, due to the poorly characterised target sequence/structure features of RNaseIII, so this question would require detailed biochemical and/or transcriptome analysis.

March 15, 2023

RE: Life Science Alliance Manuscript #LSA-2023-01954-TR

Dr. Neil Dixon
University of Manchester
Manchester Institute of Biotechnology
School of Chemistry
131 Princess Street
Manchester M1 7DN
United Kingdom

Dear Dr. Dixon,

Thank you for submitting your revised manuscript entitled "arfA antisense RNA regulates MscL excretory activity". We would be happy to publish your paper in Life Science Alliance pending final revisions necessary to meet our formatting guidelines.

A. FINAL FILES:

B. MANUSCRIPT ORGANIZATION AND FORMATTING:

****The license to publish form must be signed before your manuscript can be sent to production. A link to the electronic license to**

publish form will be sent to the corresponding author only. Please take a moment to check your funder requirements.**

Sincerely,

Reviewer #2 (Comments to the Authors (Required)):

The suggestions of the reviewer were well used to improve the manuscript. From my point of view the revised manuscript should be published.

March 20, 2023

RE: Life Science Alliance Manuscript #LSA-2023-01954-TRR

Dr. Neil Dixon
University of Manchester
Manchester Institute of Biotechnology
School of Chemistry
131 Princess Street
Manchester M1 7DN
United Kingdom

Dear Dr. Dixon,

Thank you for submitting your Research Article entitled "arfA antisense RNA regulates MscL excretory activity". It is a pleasure to let you know that your manuscript is now accepted for publication in Life Science Alliance. Congratulations on this interesting work.

DISTRIBUTION OF MATERIALS:

Again, congratulations on a very nice paper. I hope you found the review process to be constructive and are pleased with how the manuscript was handled editorially. We look forward to future exciting submissions from your lab.

Sincerely,
